# MAVOS-DD: Multilingual Audio-Video Open-Set Deepfake Detection Benchmark

## Abstract

We present the first large-scale open-set benchmark for multilingual audio-video deepfake detection. Our dataset comprises over 300 hours of real and fake videos across eight languages, with $58\%$ of data being generated. For each language, the fake videos are generated with several distinct audio and video deepfake generation models, selected based on the quality of the generated content. We organize the training, validation and test splits such that only a subset of the chosen generative models and languages are available during training, thus creating several challenging open-set evaluation setups. We perform experiments with various pre-trained and fine-tuned deepfake detectors proposed in recent literature. Our results show that state-of-the-art detectors are not currently able to maintain their performance levels when tested in our open-set scenarios. We publicly release our data and code at: `https://anonymous.4open.science/r/MAVOS-DD`.

## 1 Introduction

The rapid progress in image, audio and video synthesis technologies has enabled the creation of realistic visual content from textual descriptions (Croitoru et al., 2023; Ramesh et al., 2022; Saharia et al., 2022; Rombach et al., 2022; Nichol et al., 2022), as well as the convincing manipulation of people's identities (Li et al., 2020a; Chen et al., 2020; Joo et al., 2021; Nirkin et al., 2019) and expressions (Wang et al., 2024; Zheng et al., 2022; Hong et al., 2022; Chen et al., 2024a; Xu et al., 2024a;b; Tian et al., 2024). This has led to a surge of innovative applications across various industries, including marketing and film making. However, these breakthroughs have also fueled the rise of malicious uses, particularly in generating deceptive synthetic audio-visual content, commonly known as deepfakes (Croitoru et al., 2024). Alarmingly, a recent report shows that the incidence of deepfake-related fraud increased by a factor of 10 between 2022 and 2023[1]. In this landscape, the ability to reliably identify forged video material is more crucial than ever.

A significant body of research has emerged in response to the rising number of deepfake-related manipulation and fraud cases, aiming to detect manipulated content using advanced deep learning techniques, such as convolutional neural networks (Raza & Malik, 2023; Cozzolino et al., 2023; Kihal & Hamza, 2023; Ciamarra et al., 2024; Lanzino et al., 2024; Ba et al., 2024), transformers (Zhou & Lim, 2021; Oorloff et al., 2024; Salvi et al., 2023; Ilyas et al., 2023; Zhang et al., 2024; Nie et al., 2024), and hybrid approaches (Bonettini et al., 2021; Wang & Chow, 2023; Coccomini et al., 2022; Guan et al., 2022; Zheng et al., 2021; Choi et al., 2024). These methods have achieved remarkable results, often surpassing 99% accuracy on existing benchmarks (Croitoru et al., 2024), such as Celeb-DF (Li et al., 2020b) and FaceForensics++ (Rossler et al., 2019). Nevertheless, most evaluations are carried out in controlled environments where the synthetic and authentic samples in training and testing originate from the same video manipulation tools. This in-domain evaluation setup significantly inflates detection performance and fails to represent real-world conditions, where neither the manipulated technique nor the subject is known in advance.

To address this gap, we propose a new benchmark for evaluating audio-video deepfake detection models in a multilingual open-world setup. Our benchmark, MAVOS-DD, comprises over 40K fake and 35K real videos, totaling over 300 hours of video across eight languages: Arabic, English, German, Hindi, Mandarin, Romanian, Russian and Spanish. The fake samples are generated by several state-of-the-art deepfake generation methods based on different approaches: talking head (EchoMimic (Chen et al., 2024a), Memo (Zheng et al., 2024), Sonic (Ji et al., 2025)), portrait

---

[1]Sumsub Expert Roundtable: The Top KYC Trends Coming in 2024

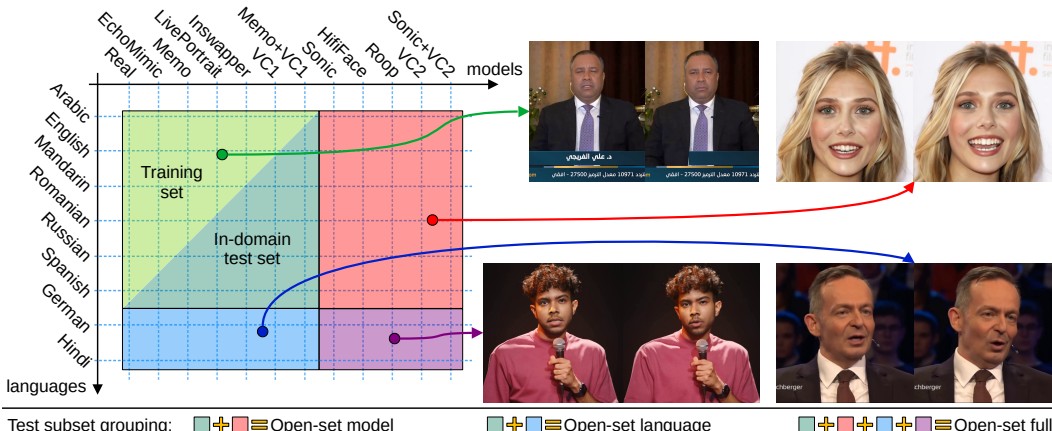

Figure 1: In MAVOS-DD, the training set and *in-domain* test set contain real and fake videos sampled from the same distribution, comprising six languages and six generative approaches. The *open-set model* test set extends the in-domain test set with fake samples generated by unseen models (Sonic, HifiFace, Roop, VC2 and Sonic+VC2). The *open-set language* test set extends the in-domain test set with samples in unseen languages (German and Hindi). The *open-set full* test set adds samples generated by unseen models in unseen languages. One fake sample from each data distribution is shown on the right-hand side. Groups of voice conversion methods (VC1 and VC2) are disjoint across all languages. Best viewed in color.

animation (LivePortrait (Guo et al., 2024)), face swap (Inswapper[2], HifiFace (Wang et al., 2021c), Roop[3]), and voice conversion (FreeVC (Li et al., 2022), KNN-VC (Baas et al., 2023), OpenVoice (Qin et al., 2024), XTTSv2 (Casanova et al., 2024), YourTTS (Casanova et al., 2022)). As shown in Figure 1, we create a multi-perspective open-set benchmark that comprises video, audio and audio-video manipulations. For audio manipulation, multiple voice conversion (VC) methods are required to cover all languages. We group VC methods in two groups, VC1 and VC2, such that methods in each group as disjoint for every language. The training set comprises audio-video samples in six languages (excluding German and Hindi), where the fake samples are generated via six approaches (excluding Sonic, HifiFace, Roop, VC2 and Sonic+VC2). We prepare an in-domain (closed) test set that is sampled from the same distribution as the training data. In addition, we create three open-set test sets: (i) *open-set model* extends the in-domain test set with fake samples generated by unseen models; (ii) *open-set language* adds German and Hindi samples to the in-domain test data; (iii) *open-set full* adds samples generated by unseen models in German and Hindi.

We perform extensive experiments using both pre-trained and fine-tuned deep fake detectors (Oorloff et al., 2024; Zou et al., 2024; Xu et al., 2023), analyzing their performance on both in-domain and open-set scenarios. While these models work well under in-domain conditions, one of them reaching an accuracy threshold of 89%, their effectiveness drops significantly in the open-set setups. The reported performance gaps highlight a critical limitation of current deepfake detection models, namely the poor generalization across deepfake generation models and languages.

In summary, our contribution is twofold:

- We present MAVOS-DD, a comprehensive multilingual open-set benchmark for audio-video deepfake detection, encompassing over 300 hours of authentic and synthetic videos across eight languages.
- We conduct a thorough evaluation of state-of-the-art deepfake detectors, uncovering substantial performance degradation when models are tested in open-world setups, thereby emphasizing the need for more robust and generalizable detection techniques.

## 2 RELATED WORK

The field of deepfake generation has seen significant advancements in recent years (Croitoru et al., 2024), particularly with the rise of diffusion models (Croitoru et al., 2023; Ho et al., 2020; Rombach

---

[2]https://github.com/deepinsight/insightface
[3]https://github.com/s0md3v/roop

Table 1: Comparison between MAVOS-DD and other video and audio-video (multimodal) datasets. MAVOS-DD is the largest dataset from multilingual audio-video open-set deepfake detection.

| Dataset | File count | | Length (h) | | | #Generative methods | #Languages | Open-set | Multimodal |
|---|---|---|---|---|---|---|---|---|---|
| | #Real | #Fake | Real | Fake | Total | | | | |
| FaceForensics++ (Rossler et al., 2019) | 1,000 | 4,000 | 4.7 | 17.0 | 21.7 | 4 | 0 | ✗ | ✗ |
| DFDC (Dolhansky et al., 2020) | 23,654 | 104,500 | 64.4 | 288.9 | 353.3 | 5 | 0 | ✗ | ✗ |
| DeeperForensics (Jiang et al., 2020) | 50,000 | 10,000 | 46.3 | 116.7 | 163.0 | 1 | 0 | ✗ | ✗ |
| ForgeryNet (He et al., 2021) | 99,630 | 121,617 | 13.3 | 13.5 | 26.8 | 15 | 0 | ✗ | ✗ |
| Celeb-DF (Li et al., 2020b) | 590 | 5,639 | 2.1 | 20.4 | 22.5 | 1 | 0 | ✗ | ✗ |
| WildDeepfake (Zi et al., 2020) | 3,805 | 3,509 | - | - | 10.9 | - | 0 | ✗ | ✗ |
| FakeAVCeleb (Khalid et al., 2021) | 500 | 19,500 | 1.1 | 41.2 | 42.3 | 3 | 1 | ✗ | ✓ |
| DeepSpeak (Barrington et al., 2024) | 6,226 | 6,799 | 17.0 | 26.0 | 44.0 | 10 | 1 | ✗ | ✓ |
| Deepfake-Eval-2024 (Chandra et al., 2025) | 1,072 | 964 | 28.9 | 16.2 | 45.1 | - | 49 | ✗ | ✓ |
| PolyGlotFake (Hou et al., 2024) | 766 | 14,472 | 2.6 | 48.3 | 50.9 | 10 | 7 | ✗ | ✓ |
| MAVOS-DD (ours) | 35,557 | 40,742 | 127.9 | 175.7 | 303.6 | 11 | 8 | ✓ | ✓ |

et al., 2022; Saharia et al., 2022; Song & Ermon, 2019). In parallel, considerable research has been devoted to developing effective detection techniques (Croitoru et al., 2024; Oorloff et al., 2024; Zou et al., 2024; Xu et al., 2023) to counter the negative effects of deepfake media. In addition, substantial efforts have been made to construct datasets for deepfake detection (Rossler et al., 2019; Dolhansky et al., 2020; Jiang et al., 2020; Li et al., 2020b; Khalid et al., 2021), thereby facilitating research in this domain.

**Audio-visual deepfake detection.** Traditional deepfake detection methods are unimodal, focusing solely on either visual artifacts, e.g. abnormal facial textures (Lanzino et al., 2024; Kingra et al., 2022; Fang et al., 2025) and inconsistent lighting (Gerstner & Farid, 2022), or audio inconsistencies, e.g. speech prosody (Blue et al., 2022; Wang et al., 2023; Attorresi et al., 2023), frequency patterns (Sriskandaraja et al., 2016; Yang et al., 2020; Fan et al., 2023; Xue et al., 2022), and voice cloning artifacts (Martín-Doñas & Álvarez, 2023; Gao et al., 2021). With generation methods becoming more capable, it is essential to leverage both visual and auditory modalities to improve the robustness and reliability of the forgery detection models (Oorloff et al., 2024; Zou et al., 2024; Xu et al., 2023). Aside from unimodal cues, utilizing multimodal (audio-visual) information can naturally capitalize on the misalignment between the two modalities by examining if the audio and video signals are coherent and temporally aligned, e.g. in terms of lip movements (Agarwal et al., 2020; Zhou & Lim, 2021) or facial expressions (Haliassos et al., 2022).

Early works on audio-visual deepfake detection used convolutional architectures (Raza & Malik, 2023; Cozzolino et al., 2023; Kihal & Hamza, 2023). For example, Multimodaltrace (Raza & Malik, 2023) extracts separate features from audio and video with residual blocks, fuses the resulting representations and further processes them to make the final prediction. Kihal & Hamza (2023) also employ individual CNN-based feature extractors, but use a Random Forest model to predict the final label. Recent works opted for architectures that leverage transformers, not only because of their higher performance, but also because of the inherent mechanism that enables fusing the information from two modalities using cross-attention modules (Zhou & Lim, 2021; Oorloff et al., 2024; Salvi et al., 2023; Ilyas et al., 2023; Zhang et al., 2024; Nie et al., 2024). Zhou & Lim (2021) detect inconsistencies between the two modalities (focusing on lip movements and speech) by aligning their low-level latent representations and fusing them through a cross-modal attention mechanism. Nie et al. (2024) employ two pre-trained frozen ViTs (Dosovitskiy et al., 2021) to extract features, with only the `[CLS]` tokens being used for classification. To bridge the gap between modalities, the audio information is integrated into the visual tokens using an audio-distilled cross-modal interaction module. Furthermore, the authors try to detect high-frequency forgery artifacts by biasing the queries, keys, and values with learnable parameters.

**Audio-visual deepfake datasets.** While the advancement of deepfake generation methods has led to the development of detection methods to defend against deepfakes, it has also driven the need for extensive datasets. In the beginning, datasets comprising data from a single modality were created for both visual (image and video) data (Dang et al., 2020; Dolhansky et al., 2020; He et al., 2021; Chen

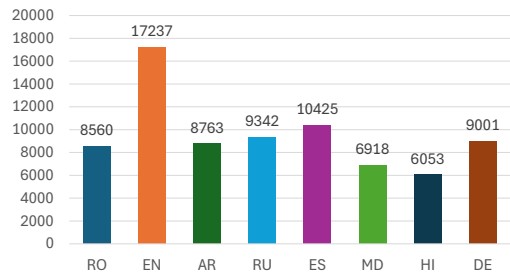
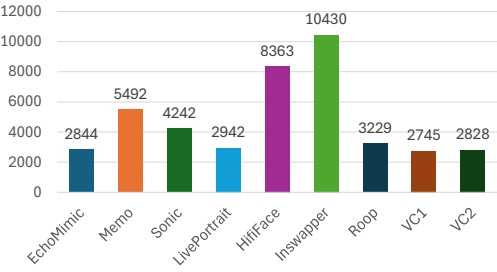

(a) Number of real and deepfake videos per language.

(b) Number of deepfake videos generated with each method.

Figure 2: Distribution of videos per language and per generative method. MAVOS-DD comprises videos in eight languages, generated with various methods. The languages are coded as follows: Arabic (AR), English (EN), German (DE), Hindi (HI), Mandarin (MD), Romanian (RO), Russian (RU) and Spanish (ES).

et al., 2024b; Rossler et al., 2019; Li et al., 2020a;b; Zi et al., 2020) and audio data (Wang et al., 2020; Liu et al., 2023). Nevertheless, with the rise of multimodal models, the availability of audio-visual datasets (Korshunov & Marcel, 2018; Khalid et al., 2021; Chandra et al., 2025; Barrington et al., 2024) has become essential.

We present a comprehensive comparison of MAVOS-DD with other video and multimodal datasets in Table 1. DFDC (Dolhansky et al., 2020) is among the largest video dataset for deepfake detection. However, multimodal datasets, such as FakeAVCeleb (Khalid et al., 2021) and Deepfake-Eval-2024 (Chandra et al., 2025) are not as large. FakeAVCeleb (Khalid et al., 2021) is based on two face swapping methods and a facial reenactment method for their synthetic English-speaking videos. While DeepSpeak (Barrington et al., 2024) tries to excel by employing 10 generative methods, Deepfake-Eval-2024 (Chandra et al., 2025) stands out by having videos in 49 languages, although 80% is English.

One of the main limitations of the deepfake detection methods is their ability to generalize to synthetic samples generated with different methods. To this end, MAVOD-DD contains samples obtained with a variety of generative methods to facilitate training robust detection models, but also to thoroughly evaluate their ability to generalize to unseen methods. Moreover, with only two exceptions (Chandra et al., 2025; Hou et al., 2024) from concurrent literature, existing datasets do not focus on the multilingual aspect of audio-visual content. Chandra et al. (2025) collect the dataset from the web, so there is no control over the generative methods and languages. In contrast, our dataset enables an open-set evaluation in terms of both generative models and languages. Furthermore, our dataset comprises $11\times$ more deepfake content (176 hours vs. 16 hours), which enables the training of very deep models with higher generalization capacity. Although their videos span 49 languages, 80% of all videos are in English (each other language representing less than 0.5% of the dataset). In this regard, MAVOS-DD provides a more even distribution across languages (see Fig. 2a). PolyGlotFake (Hou et al., 2024) contains 766 real and 14,472 fake videos, resulting in an imbalance ratio of approximately 1:18.5. In contrast, MAVOS-DD yields a more balanced ratio of roughly 1:1.3. Moreover, PolyGlotFake employs only two video manipulation methods, with the most recent being VideoRetalking (Cheng et al., 2022). In MAVOS-DD, we incorporate more recent and diverse manipulation techniques, including Sonic (Ji et al., 2025) and LivePortrait (Guo et al., 2024), offering a more up-to-date and challenging benchmark. Overall, the comparison in Table 1 shows that MAVOS-DD is the largest dataset from multilingual audio-video open-set deepfake detection.

## 3 DATASET

**Overview.** Our main contribution is MAVOS-DD, a large-scale deepfake dataset consisting of 76,299 real and synthetic videos, totaling 303 hours of content across eight different languages. The synthetic content is generated using seven video methods (EchoMimic (Chen et al., 2024a), Memo Zheng et al. (2024), Sonic (Ji et al., 2025), LivePortrait (Guo et al., 2024), Inswapper, HifiFace (Wang et al., 2021c), and Roop) and five audio methods (FreeVC (Li et al., 2022), KNN-VC (Baas et al., 2023), OpenVoice (Qin et al., 2024), XTTSv2 (Casanova et al., 2024), YourTTS (Casanova et al., 2022)). The VC methods are divided into two groups, denoted as VC1 (FreeVC, XTTSv2) and VC2 (OpenVoice, YourTTS, KNN-VC). The deepfake methods cover five key generative tasks:

Table 2: Number of real and fake videos included in the training, validation and test splits of MAVOS-DD. The test data is divided into four subsets, which generate an in-domain evaluation scenario and three open-set evaluation scenarios. The core set includes six languages (Arabic, English, Mandarin, Romanian, Russian, Spanish) and four methods (EchoMimic, Memo, LivePortrait, Inswapper). The extra languages are German and Hindi. The extra models are Sonic, HifiFace and Roop. The length (in hours) of the real and fake content in each split is reported in the last column.

| Split | | Video type | Core set | File count Extra languages | Extra models | Extra models & languages | Total count | Total length (h) |
|---|---|---|---|---|---|---|---|---|
| Train | | Real | 10,297 | 0 | 0 | 0 | 10,297 | 38.5 |
| | | Fake | 11,073 | 0 | 0 | 0 | 11,073 | 49.1 |
| Validation | | Real | 1,715 | 0 | 0 | 0 | 1,715 | 6.4 |
| | | Fake | 2,180 | 0 | 0 | 0 | 2,180 | 9.5 |
| Test | In-domain | Real | 5,185 | 0 | 0 | 0 | 5,185 | 19.3 |
| | | Fake | 5,347 | 0 | 0 | 0 | 5,347 | 25.0 |
| | Open-set language | Real | 5,185 | 7,998 | 0 | 0 | 13,183 | 46.4 |
| | | Fake | 5,347 | 4,708 | 0 | 0 | 10,055 | 49.3 |
| | Open-set model | Real | 5,185 | 0 | 10,362 | 0 | 15,547 | 56.1 |
| | | Fake | 5,347 | 0 | 15,086 | 0 | 20,433 | 78.2 |
| | Open-set full | Real | 5,185 | 7,998 | 10,362 | 0 | 23,545 | 83.1 |
| | | Fake | 5,347 | 4,708 | 15,086 | 2,348 | 27,489 | 117.1 |

talking-head generation (EchoMimic, Memo, Sonic), facial expression transfer (LivePortrait), face swapping (Inswapper, HifiFace, Roop), voice conversion (FreeVC, KNN-VC, OpenVoice, XTTSv2, YourTTS), and joint talking-head and voice conversion (Memo+VC1, Sonic+VC2). This coverage ensures a diverse and realistic set of generated videos, comprising three kinds of deepfakes: video, audio, and multimodal (audio-video). The main reason for using recent generative methods is to create a challenging dataset. Yet, another level of complexity is added through the fact that the audio-video samples cover eight languages: Arabic (AR), English (EN), German (DE), Hindi (HI), Mandarin (MD), Romanian (RO), Russian (RU) and Spanish (ES). We present the video distribution per language and per generative method in Figure 2a and Figure 2b, respectively. Note that real videos are naturally included in the distribution of videos per language, but not in the distribution of videos per generative method. The distribution per language is influenced by the number of real videos that we were able to collect for each language, while the distribution per method is influenced by the speed of each generative method. The total time required to generate all videos included in MAVOS-DD amounts to roughly 92 days (time measured on a computer with an Intel i9-14900K CPU with 192 GB of RAM and an Nvidia RTX 4090 GPU with 24 GB of VRAM).

We define official training, validation, and test splits for various evaluation scenarios, as illustrated in Figure 1. The first scenario, referred to as *in-domain* evaluation, uses a test set comprising the same languages and generative methods as the training set. The second and third scenarios, namely *open-set model* and *open-set language*, expand the in-domain test set to include samples generated by unseen models or unseen languages, respectively. The final scenario, called *open-set full*, includes samples generated by unseen models in unseen languages, presenting the most challenging evaluation setting. We present detailed statistics about MAVOS-DD and its splits in Table 2. The training and validation splits do not include videos in German or Hindi, as these languages are reserved exclusively for the test set to support open-set evaluation. Overall, the number of real and fake samples is relatively balanced. However, the *open-set model* and *open-set full* splits contain a slightly larger number of fake samples, as they comprise synthesized videos from three additional generative methods that are not present in the training set, as illustrated in Figure 1.

**Real videos.** We collect real videos from YouTube, primarily sourcing content from popular news channels or street interviews in each target language (such as EasyLanguages[4]) Additionally, we include videos from well-known channels specific to each country and language, although these are not our primary focus, as they tend to lack the diversity of speaker identities found in news broadcasts. After downloading, we apply the TalkNet active speaker detection model (Tao et al., 2021) to segment

---

[4]https://www.easy-languages.org/

the videos into shorter clips, each featuring a single speaking individual. As the process to acquire the videos and split them into smaller videos is automatic, there are some instances where the videos do not contain any humans, i.e. faces. In order to filter these out, for each video, we apply a face detector (Jocher et al., 2023) on individual frames (using a step of 15 frames) and eliminate those videos that do not have a face for more than half of the evaluated frames. The final dataset comprises 35,557 high-quality videos, with resolutions ranging from $256 \times 256$ to $1920 \times 1080$, amounting to a total of 128 hours of real content.

**Deepfake videos.** Deepfake generation typically involves a source identity image, representing the face that is manipulated by the generative model. We take these identities from multiple sources in our experiments. The first source is a set of 500 portraits generated by us using FLUX[5]. We use the simple text prompt "A portrait of a man/woman", as it consistently produces high-quality images without compromising output diversity. For the diffusion process, we set the number of denoising steps to 50 and use a guidance scale of 3.5. Additionally, we supplement the generated portraits with real identities from well-established face datasets, specifically FFHQ (Karras et al., 2021) and CelebAMask-HQ (Lee et al., 2020), along with identities found in our real videos. These datasets have disproportional dimensions, but we sample subsets from each to ensure an almost uniform distribution across datasets.

The talking-head generation is performed with EchoMimic, Memo and Sonic. We provide these models with a portrait image, sampled from the previously described set, and an audio signal containing a person speaking. The audio also originates from the real video set described earlier. The result is a video in which the person from the portrait image utters the speech from the audio file. We emphasize that the models not only manage lip synchronization, but also effectively generate head movements and facial expressions required for this task. Furthermore, we observe that Memo and Sonic perform consistently well across multiple languages, while EchoMimic struggles with languages other than English and Mandarin. For this reason, we individually fine-tune EchoMimic on additional languages, such as Romanian and Arabic, before using it for generation. We use 1,000 real videos for each language and trained the model for 10 epochs. Finally, we synthesize over 10,000 videos using talking-head generation methods, resulting in more than 65 hours of fake content. All videos are generated at a consistent resolution of $512 \times 512$ pixels.

For facial expression manipulation, we employ LivePortrait (Guo et al., 2024). This model can transfer facial movements (eyes, lips, and expressions) from a driving video to a source image or video. However, we observe a noticeable drop in quality when the person in the driving video is not directly facing the camera. Additionally, while lip synchronization is handled effectively, the transfer of eye movements and facial expressions is less effective. To address these limitations, we restrict our use to front-facing driving videos and focus only on lip synchronization. As a result, only the movements of the lips are synthesized in the generated samples, while all other facial attributes in the source video remain unchanged. The audio of the resulting video is taken from the driving video, to ensure alignment between the lips and the information spoken in the audio. We select front-facing driving videos from the set generated using talking-head synthesis, as these are primarily created from portrait images, and verified for the front-facing property. The source videos are represented by the real videos collected from YouTube. We generate over 2,900 videos using this method, resulting in more than 14 hours of fake content. The generated videos inherit the resolution of the source (real) videos, as the only changed aspect is the movement of the lips.

The face swapping is performed with Inswapper, HifiFace and Roop. Face swapping works by pasting the identity from a source image to a target video, while keeping the attributes that are not specific to the identity (facial expression, lip movement) unchanged. For the source images, we use portraits from the previously described dataset, which includes both synthetic and real identities. The target videos are selected from the collected set of real YouTube videos. Following face swapping, we apply GFPGAN (Wang et al., 2021b) for face restoration to enhance visual quality. We generate over 22,000 videos using this deepfake method, totaling 81 hours of fake content. The resolution of the resulting videos matches that of the target (real) videos.

We generate over 5,500 samples to cover both (fake audio, real video) and (fake audio, fake video) pairs. To create (fake audio, real video) samples, we apply several VC models to some of the collected real videos. The samples produced by models from VC2 are reserved exclusively for our open-set model evaluation, and are not included in the training or validation subsets. The reference voices used for the VC models are sourced from Common Voice (Ardila et al., 2020), M-AILABS (Solak, 2019),

---

[5]https://github.com/black-forest-labs/flux

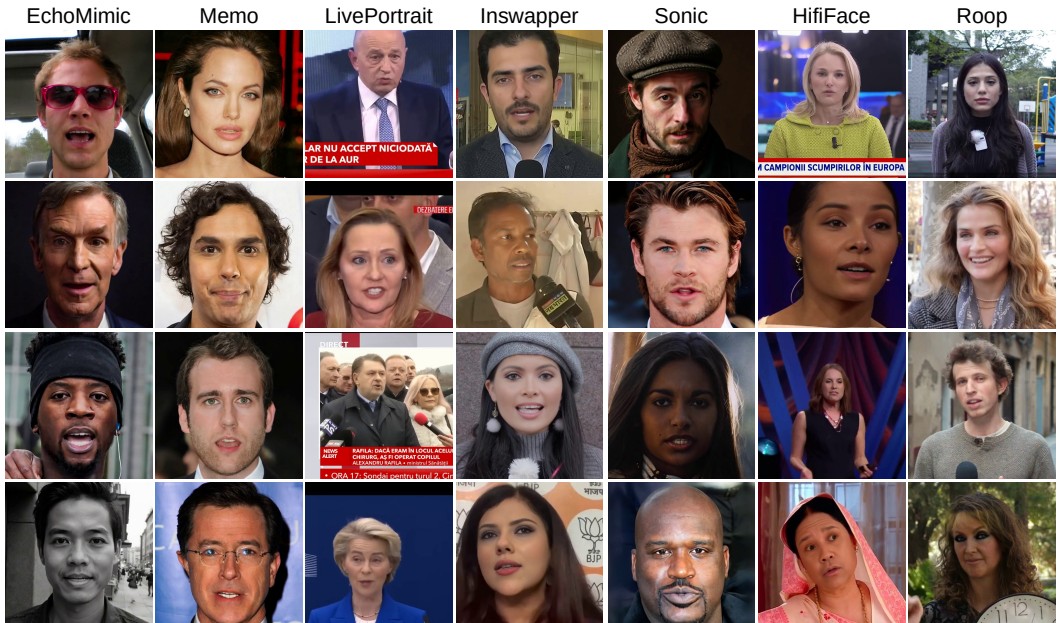

EchoMimic  Memo  LivePortrait  Inswapper  Sonic  HifiFace  Roop

Figure 3: Fake video frames generated by several deepfake generation methods. Best viewed in color.

and VoxPopuli (Wang et al., 2021a). To create the (fake audio, fake video) samples, we generate talking-head videos using Sonic and Memo, while providing fake audio samples as input.

In Figure 3, we present synthetic video frames produced by some of the deepfake methods. The samples are diverse and have a high degree of realism, confirming that MAVOS-DD represents a challenging dataset for existing deepfake detectors. For both real and generated videos, we highlight that the number of frames per second (FPS) ranges from 23 to 60. The audio bitrate varies between 88 and 140 `kbps`, with the audio sample rate spanning from 16 to 44.1 `kHz`. The video bitrate ranges from 40 to over 10,000 `kbps`.

## 4    EXPERIMENTS

**Baselines and hyperparameters.** We conduct experiments using three state-of-the-art deepfake detectors. Two of them, namely AVFF (Oorloff et al., 2024) and MRDF (Zou et al., 2024), are multimodal, while the third one, TALL (Xu et al., 2023), analyzes only the video input. AVFF employs two unimodal encoders based on transformer blocks, each of them being trained to predict features of the opposite modality. The outputs from both encoders are concatenated and passed to a binary classifier for deepfake detection. Similarly, MRDF uses two encoders to extract features from each modality. The two encoders are based on ResNet-18 (He et al., 2016). Their output is concatenated and further processed by an audio-visual transformer module for deepfake detection. TALL is a spatio-temporal modeling method that captures both spatial and temporal inconsistencies. The method is applicable to multiple architectures. In our work, we use TALL-Swin, which is based on Swin Transformer (Liu et al., 2021). We conduct the experiments using both pre-trained and fine-tuned versions of each model. We fine-tune MRDF for 5 epochs, TALL for 15 epochs and AVFF for 10 epochs on MAVOS-DD. The number of epochs are established based on early stopping. To optimize the models, we employ Adam (Kingma & Ba, 2015) with a learning rate of $10^{-3}$ for MRDF, $2 \cdot 10^{-5}$ for TALL, and $10^{-5}$ for AVFF, respectively. We keep the default values for the other hyperparameters of Adam. We set the batch size to 4 for AVFF and MRDF, and 32 for TALL. All the experiments are carried out on a computer with an Intel i9-14900K CPU with 192 GB of RAM and an Nvidia RTX 4090 GPU with 24 GB of VRAM.

**Results.** In Table 3, we report the results for the three baseline models across three evaluation metrics: mean average precision (mAP), area under the ROC curve (AUC), and accuracy (acc). We report these values on all four test sets: in-domain, open-set model, open-set language and open-set full.

The results demonstrate that MAVOS-DD is a difficult data set for existing deepfake detection methods, since all the employed and publicly available pre-trained models perform close to random

Table 3: Results obtained by pre-trained and fine-tuned versions of AVFF (Oorloff et al., 2024), MRDF (Zou et al., 2024) and TALL (Xu et al., 2023) on the MAVOS-DD official test sets: in-domain, open-set model, open-set language and open-set full. The best and second-best results on each column are highlighted in **bold blue** and orange, respectively. According to McNemar's statistical testing, all fine-tuned models are significantly better than their pre-trained counterparts (p-value < 0.001).

| Method | Fine-tuned | In-domain | | | Open-set model | | | Open-set language | | | Open-set full | | |
|---|---|---|---|---|---|---|---|---|---|---|---|---|---|
| | | mAP | AUC | acc | mAP | AUC | acc | mAP | AUC | acc | mAP | AUC | acc |
| AVFF | ✗ | 0.53 | 0.54 | 49.23 | 0.54 | 0.56 | 56.78 | 0.53 | 0.52 | 56.73 | 0.53 | 0.55 | 53.85 |
| MRDF | ✗ | 0.58 | 0.58 | 54.81 | 0.59 | 0.60 | 49.57 | 0.57 | 0.58 | 55.71 | 0.59 | 0.59 | 53.33 |
| TALL | ✗ | 0.50 | 0.50 | 49.94 | 0.47 | 0.45 | 43.60 | 0.48 | 0.48 | 49.70 | 0.47 | 0.46 | 46.38 |
| AVFF | ✓ | **0.96** | **0.96** | **89.05** | **0.91** | **0.92** | **81.74** | **0.91** | **0.91** | **85.13** | **0.90** | **0.91** | **81.64** |
| MRDF | ✓ | 0.90 | 0.91 | 83.25 | 0.76 | 0.81 | 76.42 | 0.88 | 0.89 | 82.57 | 0.80 | 0.83 | 78.36 |
| TALL | ✓ | 0.85 | 0.85 | 76.63 | 0.74 | 0.76 | 66.91 | 0.78 | 0.79 | 72.27 | 0.74 | 0.75 | 67.62 |

chance, regardless of the test set. We can attribute the performance gap of pre-trained models to the fact that MAVOS-DD typically contains examples that are more challenging to detect, since they are generated with models that exhibit a high degree of realism. The fine-tuned versions perform much better, especially in the in-domain scenario. With respect to the in-domain scenario, their performance levels decline in open-set setups, indicating that further developments are needed to improve the generalization of state-of-the-art detectors. As expected, the most significant performance drop is observed in the open-set model setup. This drop indicates that detectors still fail to generalize from a set of deepfake methods to another. The performance drop is lower in the open-set language case. However, when we examine the number of samples incorrectly predicted by the fine-tuned TALL model across in-domain and open-set language scenarios, we observe a difference of 4,597 samples, increasing from 2,457 to 7,054. This suggests that a significant portion of misclassified samples are likely mislabeled because the model only considers the video modality, thus disregarding language features. Another important observation is the noticeable performance gap between the unimodal TALL method and the two multimodal approaches (AVFF and MRDF), suggesting that jointly analyzing visual and audio modalities provides a significant advantage on MAVOS-DD.

We report the confusion matrices obtained by AVFF, MRDF and TALL, for each of the four test scenarios in Figure 4. In the open-set scenarios, AVFF shows a significant drop in its ability to detect fake videos. The same observation applies to MRDF, although the number of false negatives with respect to the in-domain test case increases by less than 6%. TALL exhibits a poor ability to detect deepfakes, regardless of the target test set. These observations strengthen the claim that MAVOS-DD represents a challenging deepfake benchmark for modern deepfake detectors. Finally, to attest the usefulness of the provided training data, we compute McNemar's statistical test between pre-trained and fine-tuned versions of each model, obtaining a p-value lower than 0.001 in all cases.

**Error analysis.** We investigate which of the deepfake generative methods poses the greatest challenge for MRDF in terms of detection accuracy. We find that samples generated by LivePortrait and Roop are the most difficult, with 80% of the samples being labeled as real. Roop is one of the methods included in the test set only, and we believe that this explains the poor performance of MRDF in identifying samples generated by Roop. In contrast, LivePortrait is part of the in-domain set, but the poor performance of the detector on this method can be attributed to the fact that we only synchronize the lips, leaving everything else as in the original video. In Figure 5, we illustrate such a scenario where we show, side-by-side, frames from a real video and its corresponding fake video modified with LivePortrait. In the illustrated video, MRDF fails to detect the fake, misclassifying it as real.

## 5 CONCLUSION AND FUTURE WORK

In this work, we introduced MAVOS-DD, a large-scale open-set benchmark for multilingual audio-video deepfake detection, comprising over 300 hours of real and generated videos. We further proposed a test split that creates four different evaluation scenarios: in-domain, open-set model, open-set language and open-set full. The resulting scenarios are aimed to assess the performance and robustness of deepfake detectors in challenging situations. We evaluated three different state-of-the-art deepfake detectors on the newly proposed benchmark, and observed significant performance

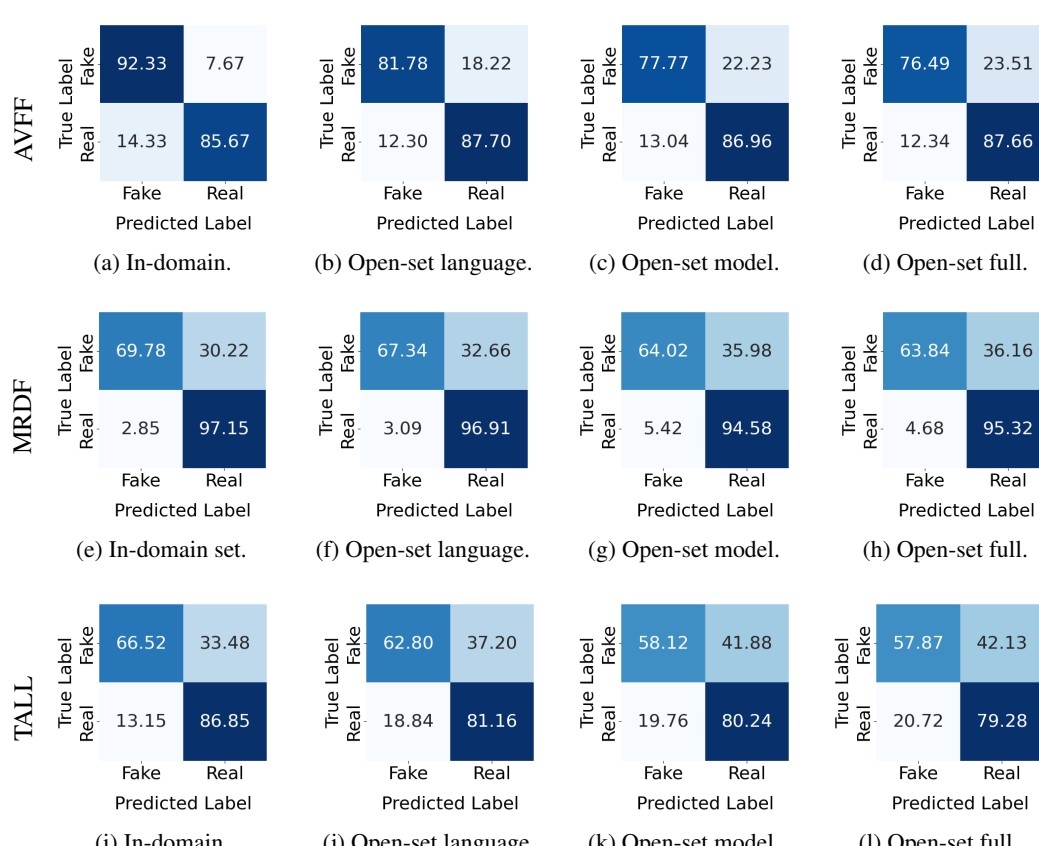

Figure 4: Confusion matrices obtained by AVFF, MRDF and TALL after fine-tuning them on MAVOS-DD.

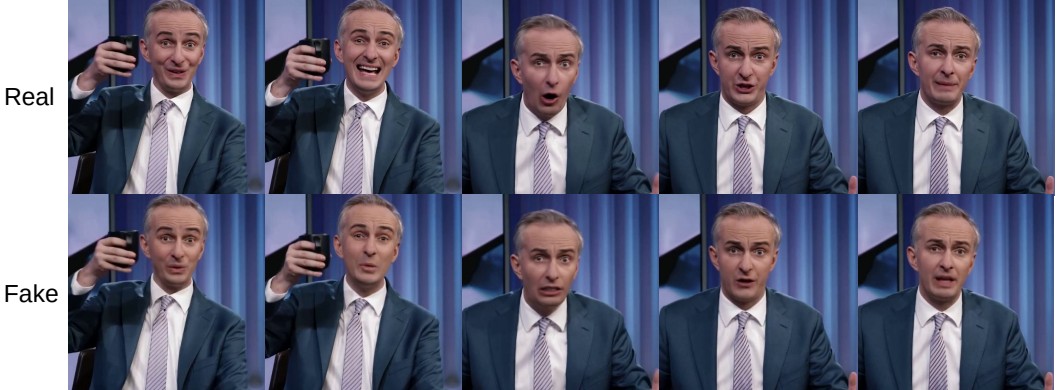

Figure 5: A real video and its corresponding fake sample generated using LivePortrait. The MRDF detector incorrectly classifies the fake sample as real. Best viewed in color.

drops across all four evaluation setups. The empirical results highlight the need to develop more robust deepfake detectors for practical scenarios.

In future work, we aim to continuously update the dataset by adding deepfake samples generated with models that are going to be released after our first release date. Thus, MAVOS-DD will keep up with the development pace of generative models, so that it will stay relevant for a long period of time. Additionally, we target the development of novel deepfake detectors that specifically address the challenges of the proposed open-set setups, which closely resemble real-world scenarios.

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

Table 4: We report language-level results with AVFF under three scenarios. In the first scenario, videos contain the original audio. In the second scenario, audio is removed by replacing it with zero values. In the third scenario, audio is randomly replaced with that of another video.

| Language | Correct Audio | | | Missing Audio | | | Mismatched Audio | | |
|---|---|---|---|---|---|---|---|---|---|
| | mAP | AUC | acc | mAP | AUC | acc | mAP | AUC | acc |
| Arabic | 0.98 | 0.98 | 94.01 | 0.69 | 0.69 | 37.81 | 0.75 | 0.77 | 64.97 |
| English | 0.93 | 0.93 | 83.31 | 0.61 | 0.62 | 48.56 | 0.64 | 0.66 | 61.53 |
| Mandarin | 0.92 | 0.94 | 86.70 | 0.70 | 0.74 | 67.26 | 0.69 | 0.74 | 69.94 |
| Romanian | 0.99 | 0.99 | 97.55 | 0.63 | 0.64 | 44.52 | 0.81 | 0.82 | 70.90 |
| Russian | 0.86 | 0.94 | 87.07 | 0.69 | 0.80 | 78.06 | 0.65 | 0.77 | 74.14 |
| Spanish | 0.89 | 0.92 | 83.77 | 0.67 | 0.70 | 63.25 | 0.67 | 0.72 | 68.49 |
| German | 0.92 | 0.92 | 87.24 | 0.68 | 0.67 | 39.65 | 0.72 | 0.73 | 62.71 |
| Hindi | 0.76 | 0.75 | 74.81 | 0.59 | 0.57 | 33.61 | 0.59 | 0.62 | 56.13 |

Yinglin Zheng, Jianmin Bao, Dong Chen, Ming Zeng, and Fang Wen. Exploring temporal coherence for more general video face forgery detection. In *Proceedings of ICCV*, pp. 15024–15034, 2021.

Yufeng Zheng, Victoria Fernández Abrevaya, Marcel C. Bühler, Xu Chen, Michael J. Black, and Otmar Hilliges. IMavatar: Implicit Morphable Head Avatars from Videos. In *Proceedings of CVPR*, pp. 13545–13555, 2022.

Yipin Zhou and Ser-Nam Lim. Joint Audio-Visual Deepfake Detection. In *Proceedings of ICCV*, pp. 14800–14809, 2021.

Bojia Zi, Minghao Chang, Jingjing Chen, Xingjun Ma, and Yu-Gang Jiang. WildDeepfake: A Challenging Real-World Dataset for Deepfake Detection. In *Proceedings of ACMMM*, pp. 2382–2390, 2020.

Heqing Zou, Meng Shen, Yuchen Hu, Chen Chen, Eng Siong Chng, and Deepu Rajan. Cross-modality and within-modality regularization for audio-visual deepfake detection. In *Proceedings of ICASSP*, pp. 4900–4904, 2024.

## A  QUALITATIVE ANALYSIS

To clarify the individual importance of audio and language features, we report language-level results. For a better assessment, we conduct two experiments. In the first setting, we remove the audio during inference by setting the audio features to zero. However, since this introduces a distribution shift between training and testing, we implement a second setting in which the audio for each video is randomly replaced with that of another video. This approach maintains a similar distribution of audio features and provides a more reliable assessment of the role played by the audio modality. We report the results for both experimental settings, along with the standard setup (where the correct audio is present) in Table 4. All experiments are conducted using the open-set language setting.

The results reported in Table 4 emphasize the impact of the audio signal on the performance of AVFF. Moreover, proper synchronization between audio and video is essential, as demonstrated by the second experimental setting, where randomly replacing the audio results in a significant performance drop. If the model was not leveraging the interaction between audio and video, we would expect little to no impact from audio replacement. However, the observed decline in performance clearly indicates that the model does rely on the audio-video cues, underscoring the importance of the audio modality.

The model was not exposed to Hindi and German during training, and its lowest performance is observed on Hindi. Given the importance of the audio modality exposed in the above experiment, its low performance on Hindi may be attributed to unique language-specific audio characteristics that the model had not previously encountered.

In addition to the above analysis, we observed no significant performance difference for the unseen languages when comparing the results on the open-set full setting to the open-set language setting. As a reminder, the key distinction between these two settings lies in the presence of fake samples

Table 5: Results for German and Hindi under open-set full and open-set language settings.

| Language | Open-set language | | | Open-set full | | |
|---|---|---|---|---|---|---|
| | mAP | AUC | Acc | mAP | AUC | Acc |
| German | 0.92 | 0.92 | 87.24 | 0.92 | 0.93 | 86.95 |
| Hindi | 0.76 | 0.75 | 74.81 | 0.76 | 0.76 | 73.12 |

Table 6: List of YouTube channels from which we downloaded the videos that we included in the dataset.

| Language | Channel List |
|---|---|
| Arabic | @aljazeera, @alhiwarchannel, @Al-Baghdadia |
| English | @TED, @ABCNews, @CNN, @aljazeeraenglish, @FoxNews |
| German | @zdfmagazinroyale, @ZDFheute, @n-tv, @tagesschau, @WELTVideoTV |
| Hindi | @timesnownavbharat, @DoordarshanNational, @ABPNews, @aupmanyu |
| Mandarin | @EasyMandarin, @setnews, @LINETODAYWORLD, @FTV_News |
| Romanian | @StirileProTV, @Antena3CNN, @digi24hd56, @catalinmoise, @StareaNatiei |
| Russian | @InRussianFromAfar, @EasyRussianVideos, @tvrain, @vvhelp |
| Spanish | @HolaSpanish, @EasySpanish, @dwespanol, @rtvenoticias |

generated using unseen generative models in the open-set full scenario. The similar performance across both settings for the unseen languages suggests that the primary challenge stems from the language-specific characteristics themselves, rather than from the generative methods. The detailed results are presented in Table 5.

Given the previously highlighted importance of the audio modality, the lower performance on unseen languages (German and Hindi) can be partially attributed to language-specific audio characteristics. However, visual cues may also contribute to the performance drop. This is visible from the results reported for TALL in the main paper. Its performance in the open-set language setting is significantly lower than on the in-domain set. Since TALL relies solely on visual information, this suggests that differences in visual features, such as mouth movements or scene variations, also play a role. Therefore, the reduced performance on unseen languages likely stems from a combination of factors, including visual, audio, and cross-modal (audio-visual) discrepancies.

## B    DATA SOURCE

In Table 6, we list the YouTube channels from which we downloaded our real videos. The list demonstrates that the real videos are collected from a variety of different sources.

## C    ETHICAL STATEMENT

We share MAVOS-DD under the International Attribution Non-Commercial Share-Alike 4.0 (CC BY-NC-SA 4.0) license, aiming for open and responsible research on deepfake detection. All real data samples are collected from public YouTube videos. Since the videos are gathered from a public website, we adhere to the European regulations[6] allowing researchers to use and store data from the public web domain for non-commercial research purposes. Moreover, we respect the individual privacy rights, including the right to be forgotten. If any individual identifies themselves in the dataset and wishes to have their data removed, they can contact us and we will promptly address the request by removing the respective video(s), in compliance with data protection principles.

## D    BROADER IMPACT AND LIMITATIONS

The advancements of deepfake generation models have significant implications for society, as it facilitates the widespread of misinformation. As synthetic media becomes increasingly realistic and

---

[6]https://eur-lex.europa.eu/eli/dir/2019/790/oj

accessible, the risk of misuse continues to grow. To fight against this, not only more competent models are required, but also varied datasets, as robust detection systems heavily depend on the utilized training data. Our research fosters the development of such models, as it addresses some of the limitations of previous datasets: a wide range of generation methods, multiple languages, and a meticulously designed split that translates into challenging open-set evaluation scenarios. Robust deepfake detection models may be beneficial for journalists, social media platforms and even governmental agencies. It could also help to protect individuals from having their reputation damaged.

Nevertheless, we also acknowledge that the development of detection methods can also lead to more sophisticated generative models, the research in the generative AI domain being restless. Still, we are convinced that MAVOS-DD will continue to be very useful, as we aim to continuously update it with state-of-the-art generative models.

A potential limitation of our benchmark consists of the hardware requirements to carry out experiments on it. Some minimum resources, e.g. CPU for loading the videos and GPU for deep learning models, must be utilized for training and evaluating on such a dataset. Another possible limitation is represented by the fact that the dataset inadvertently has a demographic bias, corresponding to the set of eight languages, which could result in reduced performance between different populations. This requires a continued evaluation of fairness and increased responsibility when deploying deepfake models trained on our dataset.

