# OpenReview forum: "MAVOS-DD: Multilingual Audio-Video Open-Set Deepfake Detection Benchmark"
_ICLR.cc/2026/Conference — ICLR 2026 Conference Withdrawn Submission_

### Official Review · Reviewer_8KxZ · 2025-10-29

**Soundness:** 3
**Presentation:** 3
**Contribution:** 3
**Rating:** 6
**Confidence:** 3

**Summary:**

This paper introduces and releases MAVOS-DD, a novel, large-scale benchmark dataset for deepfake detection. The core contribution of this dataset lies in its multilingual and open-set characteristics, specifically designed to address the critical limitation of current detection models—their poor generalization when confronted with unseen generation methods and unknown languages. MAVOS-DD comprises over 300 hours of audio-visual data spanning eight languages, synthesized using a variety of advanced forgery techniques, including video-based methods (e.g., face reenactment, face swapping) and audio-based methods (e.g., voice conversion).
The authors meticulously define four evaluation scenarios: in-domain, open-set model, open-set language, and fully open-set (open-set full). By evaluating three existing state-of-the-art (SOTA) detection models—AVFF, MRDF, and TALL—on MAVOS-DD, the experimental results clearly demonstrate a significant, and sometimes dramatic, performance drop when these models are transferred from in-domain to open-set settings.
Through these findings, the paper highlights the limitations of current deepfake detection technologies in real-world scenarios and underscores the urgent need to develop more robust and generalizable detectors.

**Strengths:**

This paper precisely identifies a core challenge in deepfake detection—model generalization. Most existing studies evaluate under closed-set assumptions, leading to inflated performance metrics that fail to reflect how models would perform in real-world, unpredictable environments.
By introducing two key variables—"unseen generation models" and "unseen languages"—MAVOS-DD provides the research community with a highly valuable evaluation benchmark that more closely mirrors real-world challenges.
One of the paper’s most notable strengths is its clear and rigorous experimental design. Partitioning the test set into four progressively more challenging subsets—in-domain, open-set model, open-set language, and open-set full—is an insightful approach.
This not only establishes a unified evaluation protocol for future work on this dataset but also enables researchers to quantitatively disentangle performance degradation along different generalization dimensions (e.g., to novel generation models vs. novel languages), providing a powerful tool for diagnosing model weaknesses.

**Weaknesses:**

The authors mention fine-tuning the EchoMimic model to adapt it to new languages, but this is only briefly stated as “using 1,000 videos and training for 10 epochs.” Crucially missing are details about the network architecture, loss function, optimizer, learning rate, and other key hyperparameters used during fine-tuning. This lack of information prevents others from reproducing samples with the same artifact characteristics.
Due to quality concerns, the authors imposed strict constraints on their use of the LivePortrait model (e.g., frontal faces only, lip movements only). This may introduce a specific bias characteristic of a "laboratory" setting into the generated samples.
As a result, detection models might learn to recognize this "imperfect yet specific" forgery pattern rather than the general artifacts produced by the model under typical conditions. The paper does not sufficiently discuss this potential bias and its implications for the fairness of the evaluation.

**Questions:**

There is a significant imbalance in the number of videos across different languages (e.g., approximately 17k English videos versus only about 6k German videos). Could this imbalance affect the training and evaluation of multilingual models? Did the authors employ language-stratified sampling when splitting the training and test sets?
Splitting the voice conversion (VC) models into VC1 and VC2, and specifically reserving VC2 for open-set testing, is a well-considered design choice. What criteria were used to decide which specific models were assigned to VC1 versus VC2?
The unimodal TALL model performs significantly worse than multimodal models across all scenarios. Beyond the obvious reason—“ignoring audio information”—does this suggest that the visual artifacts in MAVOS-DD are inherently more subtle or harder to detect than those in other datasets like FakeAVCeleb? Or could it indicate that the visual artifacts in MAVOS-DD are highly correlated with audio artifacts, necessitating joint analysis for effective detection?

**Details Of Ethics Concerns:**

The authors deserve credit for thoroughly addressing ethical considerations and providing clear statements in Sections C and D of the appendix. They explicitly note that all real videos are sourced from publicly available YouTube channels and comply with relevant European regulations. Additionally, they provide a contact channel for data removal, demonstrating respect for individual privacy rights—a responsible and commendable practice.

---

### Official Review · Reviewer_6VGA · 2025-10-30

**Soundness:** 2
**Presentation:** 2
**Contribution:** 2
**Rating:** 2
**Confidence:** 4

**Summary:**

The paper introduces MAVOS-DD which is a new benchmark dataset for audio-visual deepfake detection. The dataset is multilingual (8 languages) and explicitly open-set (certain languages and generative models are withheld during training). Fake videos are created using a variety of video-generation models and audio-generation methods. The authors create in-domain and three open-set cases for proper evaluation. The paper claims that there is a generalization gap in current detectors through evaluation benchmarks in the open-set performance.

**Strengths:**

1. The dataset is designed using open-set splits, which mimics real-world deployment scenarios.
2. The created dataset is comprehensive, encompassing more than 300 hours in length and 8 languages.
3. Utilizes recent deepfake mechanisms in a targeted manner to make comprehensive deepfake examples.

**Weaknesses:**

1. The benchmark is conducted on three models (two multimodal and one video only). This is fairely limited evaluation, and there should have been more benchmarking experiments. Some relevant papers are [1][2][3]. Furthermore, the scores in the in-domain set and open-set full sets reach 0.96 mAP and 0.9 mAP respectively. So how challenging is this dataset compared to existing datasets, and what is the exact nature of these challenges?

2. The novelty contributions of the dataset is limited to the design of an open-set split. The dataset is a scaled-up version of existing works, and multilingual audio-visual deepfake datasets already exist. Thus, I think the novelty is a bit limited in terms of unique contributions.

3. The ablation study could have been more robust regarding the characteristics of the dataset. Such as ablation on finegrained video event and their diversity, distribution of video lengths, etc. There is no cross-dataset evaluation performed, where a model trained on MAVOS-DD is evaluated on another.

4. Human evaluation of such datasets is needed, to compare against the benchmark results. No such evaluation is provided.

5. The real videos are taken from a limited number of YouTube channels. This can result in the real-set being very biased towards specific contents.

[1] Yang, Yongqi, et al. "D^ 3: Scaling Up Deepfake Detection by Learning from Discrepancy." Proceedings of the Computer Vision and Pattern Recognition Conference. 2025.
[2] Bohacek, Matyas, and Hany Farid. "Lost in translation: Lip-sync deepfake detection from audio-video mismatch." Proceedings of the IEEE/CVF Conference on Computer Vision and Pattern Recognition. 2024.
[3] Han, Yue-Hua, et al. "Towards More General Video-based Deepfake Detection through Facial Component Guided Adaptation for Foundation Model." Proceedings of the Computer Vision and Pattern Recognition Conference. 2025.

**Questions:**

1. Did the authors do experiments to compare against other benchmark datasets? What unique benefit does this dataset provide compared to existing datasets?
2. How would a model trained on MAVOS-DD perform on existing datasets in a zero-shot setting?
3. Do detectors fail more on novel appearance cues (visual artifacts) or novel audio (voice differences)? Is there any qualitative analysis done on the benchmarks?
4. Is there any human validation of the dataset created? Is human performance evaluated as well?

**Details Of Ethics Concerns:**

As the dataset consists of YouTube and deepfake videos, the authors need to be legally compliant before releasing the dataset. The paper does have an ethical statement, but this needs to be reviewed properly.

---

### Official Review · Reviewer_uJxp · 2025-10-30

**Soundness:** 2
**Presentation:** 3
**Contribution:** 3
**Rating:** 4
**Confidence:** 3

**Summary:**

This paper presents MAVOS-DD, a large-scale multilingual audio-video open-set deepfake detection benchmark.
It contains 76K real and fake videos (≈300 h) across 8 languages and multiple generation models.
The benchmark introduces in-domain and open-set splits (unseen models, unseen languages, combined) to test detector generalization.
Experiments on three detectors (AVFF, MRDF, TALL) show strong in-domain accuracy but large performance drops in open-set settings, revealing poor cross-model and cross-language robustness.

**Strengths:**

-First truly multilingual open-set benchmark for multimodal deepfake detection.
-Large-scale, balanced dataset covering diverse generation methods.
-Clear experimental protocol and significant empirical findings on generalization limits.
-Public release of data and code promotes reproducibility.

**Weaknesses:**

A notable limitation of the paper is the underexplored multilingual aspect.
Although MAVOS-DD includes eight languages, the experiments do not analyze performance across different linguistic groups or phonetic structures. There is no discussion of how language-related features (e.g., tonal versus non-tonal prosody, articulation speed, or lip movement diversity) might affect audio–visual coherence and thus detection difficulty.
As a result, the “multilingual” claim feels primarily structural rather than analytical, missing an opportunity to connect linguistic diversity with multimodal forgery realism.
Another weakness is lack of deeper analysis of multimodal inconsistencies between audio and video.
While the dataset is explicitly designed for audio–visual deepfake detection, the paper merely reports overall accuracy drops without exploring why the models fail. There is no investigation of which cross-modal cues—such as lip–speech synchronization, facial expression–tone consistency, or temporal alignment—are most responsible for detection errors.
This omission weakens the interpretability and scientific insight of the work; a more systematic error or ablation analysis could reveal valuable findings about the limits of current multimodal detectors.

**Questions:**

Have the authors conducted any language-level analysis or pilot experiments to examine how linguistic characteristics influence audio–visual deepfake detection?
For instance, do tonal languages (like Mandarin) or languages with strong coarticulation effects lead to higher lip–speech synchronization errors or pose greater challenges for multimodal detectors?
Even a brief discussion or preliminary observation on these cross-linguistic effects would substantially strengthen the paper’s multilingual contribution.

---

### Official Review · Reviewer_7P1s · 2025-10-30

**Soundness:** 3
**Presentation:** 3
**Contribution:** 3
**Rating:** 4
**Confidence:** 4

**Summary:**

This work introduces MAVOS-DD, the first audio-visual deepfake detection benchmark designed for multilingual and open-set scenarios, featuring 300 hours of data across eight languages, diverse generator combinations, and open-set evaluation splits.

**Strengths:**

The paper demonstrates that current audio-visual deepfake detection methods still exhibit insufficient generalization in open-world scenarios, which is a particularly important and often overlooked issue.

**Weaknesses:**

1 The definition of “open-set” essentially refers to domain mismatch—that is, generalization. The key point is how to generalize to unseen attacks, unseen languages, etc. Therefore, the authors should emphasize this aspect more clearly. Using the term “open-set” alone may mislead readers into thinking it refers to environmental variations in real-world testing.

2 The paper only verifies its claims through performance degradation in experiments, but lacks deeper analysis. For example, can the authors investigate which types of attacks fail to generalize? What is the cause of language-specific non-generalization—is it due to linguistic characteristics or limitations of current models?

3 Although this is primarily a benchmark work, beyond collecting data, the authors could still provide deeper analysis and propose potential generalization/adaptation strategies, or at least meaningful discussion.

4 It would be beneficial to detail the full data construction pipeline—especially what automated scripts were used and what processing steps the data went through, as these decisions significantly affect data distribution.

**Questions:**

See the above weaknesses

---

### Official Review · Reviewer_V3Sy · 2025-11-01

**Soundness:** 3
**Presentation:** 3
**Contribution:** 3
**Rating:** 4
**Confidence:** 3

**Summary:**

This paper presents MAVOS-DD, a large scale multilingual benchmark for open set audio video deepfake detection. The dataset contains more than 300 hours of real and synthetic videos in eight languages, covering several major manipulation types including talking head animation, face swap, expression reenactment, and voice conversion. The authors define four evaluation settings, namely in domain, open set model, open set language, and open set full, to measure generalization under unseen generative models and unseen languages.

**Strengths:**

1. MAVOS-DD is a benchmark to offer explicitly defined training, validation, and four testing splits that jointly vary across languages and generative models.
2. The dataset spans eight languages with relatively balanced distributions and integrates both visual and auditory manipulations, making it broader than previous resources such as FakeAVCeleb or PolyGlotFake.
3. The authors conduct detailed analyses including model ablations, language wise results, and audio video synchronization tests, which demonstrate the difficulty of the proposed benchmark.

**Weaknesses:**

1. The "multilingual" aspect, as a key novelty, is an incremental contribution given the emergence of other benchmarks.
2. The evaluation is entirely self-contained within MAVOS-DD, and all results are based on models fine-tuned and tested on the proposed dataset. The paper does not examine how the fine-tuned models perform on existing benchmarks such as Deepfake-Eval-2024 or PolyGlotFake. Such cross-dataset validation would provide a broader form of open-set evaluation, verifying whether the proposed splits capture generalization patterns that also hold across different datasets. This analysis would strengthen the claim that MAVOS-DD reflects realistic open-world conditions rather than dataset-specific separation. Moreover, no human baseline is reported. Measuring human accuracy on a subset of the test data could provide useful evidence of dataset difficulty and help calibrate the claimed challenge level of the benchmark.
3. The dataset mainly targets modification-type forgeries such as face swap, voice conversion, and talking head animation, rather than fully generative audio-video synthesis. With the rapid emergence of text-to-AV diffusion systems such as Sora, this restriction may reduce the long-term relevance of the benchmark.

**Questions:**

Please also refer to the issues discussed in the Weaknesses section.

---

### Note · Authors · 2025-11-12

I have read and agree with the venue's withdrawal policy on behalf of myself and my co-authors.